# Antiviral Molecular Targets of Essential Oils against SARS-CoV-2: A Systematic Review

Muhammad Iqhrammullah [1,2] , Diva Rayyan Rizki [2,3,*,†] , Agnia Purnama [2,*,†] , Teuku Fais Duta [3] , Harapan Harapan [3,4,5] , Rinaldi Idroes [6,7,8] and Binawati Ginting [6,*]

1    Faculty of Public Health, Universitas Muhammadiyah Aceh, Banda Aceh 23245, Indonesia
2    Innovative Sustainability Lab, PT. Biham Riset dan Edukasi, Banda Aceh 23243, Indonesia
3    Medical Research Unit, School of Medicine, Universitas Syiah Kuala, Banda Aceh 23111, Indonesia
4    Tropical Disease Centre, School of Medicine, Universitas Syiah Kuala, Banda Aceh 23111, Indonesia
5    Department of Microbiology, School of Medicine, Universitas Syiah Kuala, Banda Aceh 23111, Indonesia
6    Department of Chemistry, Faculty of Mathematics and Natural Sciences, Universitas Syiah Kuala, Banda Aceh 23111, Indonesia
7    Department of Pharmacy, Faculty of Mathematics and Natural Sciences, Universitas Syiah Kuala, Banda Aceh 23111, Indonesia
8    Herbal Medicine Research Center, Universitas Syiah Kuala, Banda Aceh 23111, Indonesia
*    Correspondence: divrayriz@gmail.com (D.R.R.); agniapurnama2@gmail.com (A.P.); binawati@unsyiah.ac.id (B.G.)
†    These authors contributed equally to this work.

**Abstract:** Essential oils are potential therapeutics for coronavirus disease 2019 (COVID-19), in which some of the volatile compounds of essential oils have been well known for their broad antiviral activities. These therapeutic candidates have been shown to regulate the excessive secretion of pro-inflammatory cytokines, which underlies the pathogenesis of severe COVID-19. We aimed to identify molecular targets of essential oils in disrupting the cell entry and replication of SARS-CoV-2, hence being active as antivirals. Literature searches were performed on PubMed, Scopus, Scillit, and CaPlus/SciFinder (7 December 2022) with a truncated title implying the anti-SARS-CoV-2 activity of essential oil. Data were collected from the eligible studies and described narratively. Quality appraisal was performed on the included studies. A total of eight studies were included in this review; four of which used enzyme inhibition assay, one—pseudo-SARS-CoV-2 culture; two—whole SARS-CoV-2 culture; and one—ACE2-expressing cancer cells. Essential oils may prevent the SARS-CoV-2 infection by targeting its receptors on the cells (ACE2 and TMPRSS2). Menthol, 1,8-cineole, and camphor are among the volatile compounds which serve as potential ACE2 blockers. β-caryophyllene may selectively target the SARS-CoV-2 spike protein and inhibit viral entry. Other interactions with SARS-CoV-2 proteases and RdRp are observed based on molecular docking. In conclusion, essential oils could target proteins related to the SARS-CoV-2 entry and replication. Further studies with improved and uniform study designs should be carried out to optimize essential oils as COVID-19 therapies.

**Keywords:** anti-inflammation; antiviral; aromatic plant; SARS-CoV-2; volatile oil

## 1. Introduction

Since declared as a pandemic, coronaviruses disease 2019 (COVID-19) caused by severe acute respiratory syndrome coronavirus 2 (SARS-CoV-2) has been responsible for the mortality of more than 6.5 million lives as of 15 December 2022 [1]. The disease has a broad range of clinical features, from common cold symptoms (i.e., cough, headache, and fever) up to lethal complications (i.e., pneumonia, multiple organ failure, and even death). This novel positive-sense single-stranded RNA virus infects and damages the proximal airway epithelial cells, which eventually leads to pneumonia [2]. Angiotensin-converting enzyme 2 (ACE2) acts as a protector of vascular tissues, an Angiotensin II effects stabilizer, provides endothelial protection, and promotes mechanisms of regeneration. The virus

gains access to the host cell via ACE2 receptors which are highly expressed in bronchial epithelial cells of the lower respiratory tract [3]. A vaccination program has been run by the government to slow down the transmission of SARS-CoV-2, but despite its efficacy, the distribution and acceptance rate remained challenging [4,5]. Researchers continue to pursue the investigation of potential anti-SARS-CoV-2 drugs, where some have shown effectiveness against the virus but with limitations in the approval status, prescription, and administration route [6,7]. Moreover, the emergence of the Omicron variant of SARS-CoV-2 has set a new challenge in curbing this public health problem deriving from reduced efficacy in the prophylaxis and treatment [8].

Essential oils have gained attention from researchers worldwide for their potential use in COVID-19 clinical management. Essential oil is comprised of single or multiple volatile components classified as phenylpropanoids, monoterpenes, or sesquiterpenes. These mixtures of volatile compounds have been reported for their broad antiviral activities, including anti-yellow fever virus, anti-avian influenza, anti-influenza virus, anti-human immunodeficiency virus (anti-HIV), and anti-human herpesviruses [9]. Immunomodulating activities exerted by the essential oils have been reported as responsible for stimulating immune response while suppressing the impact of the inflammatory reaction induced by viral infection [10,11]. Hence, it is not surprising that clinical trials have been carried out to observe the efficacy of essential oil as a supplemental treatment in COVID-19 management [12–15]. The effects generated by essential oils on COVID-19 patients include rapid viral clearance, reduced fatigue, and reduced time to recovery [12–15].

Research on SARS-CoV-2, particularly in its molecular mechanisms and pathogenesis, has progressed rapidly [16]. The involvement of the ACE2 receptor and TMPRSS2 as the gateway for SARS-CoV-2 entry has been reported [17]. Identification and classification of SARS-CoV-2 spike protein has shed light on the viral entry mechanism [18]. During viral replication, just like most single-stranded RNA viruses, SARS-CoV-2 requires proteases and (RNA-dependent RNA polymerase) RdRp [19]. By understanding the proteins involved during the SARS-CoV-2 infection, researchers could use this information to find proper drugs for COVID-19 management [20]. Herein, we would emphasize the molecular mechanisms of essential oils as anti-SARS-CoV-2 based on published evidence through a systematic review. Indeed, there have been several reviews emphasizing the anti-SARS-CoV-2 potential of essential oils [21,22]. One of which was a narrative review [21], while another was a systematic review [22]. However, this is the first systematic review focusing on the molecular targets of essential oils while acting as anti-SARS-CoV-2. In addition, we also performed a quality appraisal analysis on the included studies using our own developed tool that is suitable for anti-SARS-CoV-2 research, adding the novelty aspect of this systematic review.

## 2. Research Question

This systematic review aimed to answer "What are molecular targets of essential oils as anti-SARS-CoV-2?". The primary extracted data included the effects of the essential oil exposures on the molecules related to the entry and replication of SARS-CoV-2 in vitro or in vivo. The secondary included data included the efficacy of essential oil in inhibiting SARS-CoV-2 and attenuating the pro-inflammatory factors.

## 3. Methods

### 3.1. Search Strategy

We searched literature indexed in four databases, namely PubMed, Scopus, Scillit, and CaPlus/SciFinder, up to 7 December 2022. Identification of the literature was carried out by following truncated title combination: 'essential oil', 'aromatic plant', 'SARS-CoV-2', 'COVID', and 'coronavirus'. Boolean operators 'OR' and 'AND' were used in all databases, resulting in the combination: 'essential oil' OR 'aromatic plant' AND 'SARS-CoV-2' OR 'COVID' OR 'coronavirus'.

### 3.2. Inclusion and Exclusion Criteria

Studies should report the molecules targeted by the essential oil administration aiming to screen anti-SARS-CoV-2. Studies with in vitro and/or in vivo research design were considered eligible. Studies that did not investigate the molecular target, combination of essential oils with other therapeutic candidates, and studies reporting molecular docking results without confirmation from in vitro or in vivo studies were all excluded. We limited the publication year to 2020 onward since the pandemic brought intensely new studies because the genomic data of SARS-CoV-2 (hence the molecular targets) started being available in early 2020 [23]. Only studies reported in English language were included. 'Grey' literature, such as conference paper, was included, but other types of documents, such as patent, editorial, commentary, or erratum were excluded.

### 3.3. Screening and Selection of the Records

We followed the Preferred Reporting Items for Systematic Reviews and Meta-Analyses (PRISMA) to report the screening and selection process. Duplicate removal was performed with the help of EndNote 19 (Clarivate Analytics, Philadelphia, PA, USA) after importing all searched records into the software. The screening was performed based on the 'title and abstract' and the full content of the manuscript, sequentially, by two independent reviewers (M.I. and A.P.). The selection was based on the eligibility criteria set in this study, where discrepancies occurred were resolved through re-checking the articles, discussion, and consultation with the third reviewer (B.G.). After obtaining the eligible studies, we screened the citing and cited studies for inclusion.

### 3.4. Data Extraction and Presentation

Molecules targeted by the essential oils, along with the activities (inhibition, down-regulation, or upregulation), were extracted from the included studies. If applicable, data indicating the effectiveness of the activity, such as selective index (SI), median inhibitory concentration ($IC_{50}$), and/or inhibition percentage, were collected. The type of essential oils, major constituents of the essential oils, and study design were also extracted from the table. Identities of each literature, such as author's name and publication year, were listed.

### 3.5. Quality Appraisal

Due to the absence of a standardized quality appraisal tool suitable for in vitro antiviral study, we developed our own metrics following the suggestion of a previously published report and studies cited therein [24]. Quality appraisal criteria used are presented and explained in Table 1. 'Yes' and 'No' indicators were used to indicate the criteria fulfillment by each study. Quality appraisal was performed by M.I. and A.P., where disagreement was resolved by re-evaluating the literature, discussion, and consultation with B.G.

**Table 1.** Criteria used to appraise the quality of the included study.

| Criteria | Description |
|---|---|
| Study design rationale | Study design is in line with the research question |
| Reproducibility | Methods are clear<br>Materials and samples are presented in detail. |
| Replication | Performed in triplicate or more |
| Negative/positive control | Results compared with positive/negative control |
| Anti-SARS-CoV-2 activity | Investigation on the viral entry or replication, regardless the types of cell culture used. |
| Study adequacy | Study design is sufficient to reveal the molecular mechanism of essential oils as anti-SARS-CoV-2 |

## 4. Results

### 4.1. Results from the Literature Search

The literature search and selection process workflow, along with the number of publications obtained from each step, are presented in Figure 1. A total of 2112 studies were identified from four databases, and the number was reduced to 1990 studies after removing the duplicates. Screening the title and abstract of each study resulted in 148 studies deemed relevant to the research question. Of these, the exclusion of other studies was carried out to 143 studies, where 48 of them were excluded because of the document type (i.e., review, patent, erratum, editorial, and commentary). As many as 14 studies were excluded from the main review because they only reported results from in silico studies without further confirmation from in vitro or in vivo analysis. Nonetheless, we made a tabulation for such studies to observe the faith of molecular docking studies; whether the investigation is continued to in vitro studies. A study investigated the activities of essential oil components (terpenes) against viral replication, RdRp expression, and spike protein expression of SARS-CoV-2, but the samples were combined with cannabidiol, hence excluded [25]. Studies ($n = 4$) that did not report molecular targets, despite the anti-SARS-CoV-2 activities, were excluded as well [26–29]. Three additional studies were obtained from screening the reference lists of the five priorly included studies [30–32].

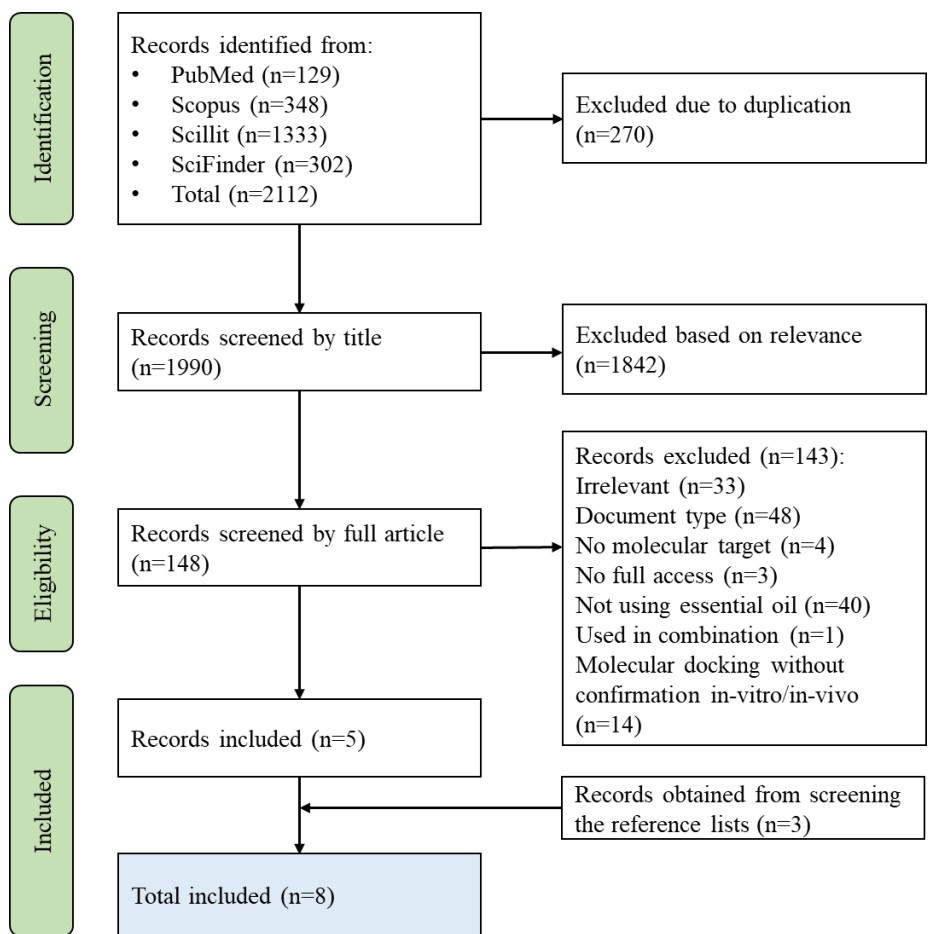

**Figure 1.** PRISMA diagram for the screening and selection process of the published literatures.

### 4.2. Characteristics of the Included Studies

Details of the included studies, including their characteristics and study outcomes presented in Table 2. A total of six studies procured their essential oils as commercial products [30–35]. Other studies obtained essential oils through hydrodistillation [36] and a combination of aqueous extraction and n-hexane partition [37]. Essential oils from

*Eucalyptus* sp. were reported in three reports [33,35,36]. Similar to *Citrus* sp., which had been reported in three studies [34,35,37]. One study covered a screening of ACE2 inhibition by 30 plant species [35]. Four studies used enzyme inhibition assays [30–33], while four other studies used various cell cultures, namely Vero-E6 cells [36,37], Huh7 cells [37], HEK293 cells [34], and HT-29 cells [35]. Only three studies evaluated the anti-SARS-CoV-2 of the essential oils [34,36,37], where one of which used the HIV-based pseudotype SARS-CoV-2 [34].

### 4.3. Primary Outcome

Primary outcomes in the form of essential activities against SARS-CoV-2-related proteins are presented in Table 2. Three studies revealed the interaction of *Eucalyptus* sp., *Mentha* sp., and *Rosmarinus officinalis* essential oils with human ACE2 based on enzyme inhibition assay [30,31,33]. Using HT-29 colorectal adenocarcinoma cell lines, a study revealed the antagonistic effects of essential oils against ACE2 expression [35]. Among 30 essential oils screened, those from *C. limon* and *P. graveolens* were the most potent in inhibiting and downregulating ACE2, which are also found to be effective in downregulating TMPRSS2 mRNA expression [35]. β-caryophyllene, a major component of many essential oils, was effective and selective against SARS-CoV-2 spike protein [34]. Intriguingly, however, the exposure of *Eucalyptus globulus* essential oil to the infected cells promoted viral entry (viral replication in the treated cell culture was higher than that in control) [34]. Essential oils from *A. robusta* bark had a selectivity index (SI) of 17.5 against SARS-CoV-2 infection based on the Vero-E6 model, where interactions of its major components (such as tricyclene, α-pinene, d-camphene, and limonene) with SARS-CoV-2 M$^{pro}$, RdRp, and RBD had been revealed using molecular docking approach [36]. Unfortunately, no further confirmation, either using in vitro or in vivo study, was carried out on the foregoing interactions [36].

### 4.4. Secondary Outcome

The secondary outcomes of this review include pro-inflammatory proteins which contribute to the COVID-19 manifestation, where the extracted data are presented in Table 2. We found at least two proteins involved in the inflammation cascade of the immune response following the SARS-CoV-2 infections. The studies based on enzyme inhibition assay suggested that essential oils from *Eucalyptus* sp., *Mentha* sp., and *R. officinalis* had inhibitory activities against 5-LOX [30,31,33]. A study witnessed increased expressions of TNF-α and IL-6 concomitant to SARS-CoV-2 infection in Huh7 cells [34]. Treatment using essential oils from *Citrus clementine* successfully downregulates the expressions of TNF-α and IL-6 [34].

### 4.5. Results from the Quality Appraisal

Quality appraisal results of the included studies have been presented in Table 3. All studies satisfy the 'study design rationale', 'reproducibility', and 'replication' criteria. A negative control is employed by all studies, but it is not the case for a positive control. The positive control was only used for pro-inflammatory proteins [30–33], and none was used for ACE2 or TMPRSS2. Anti-SARS-CoV-2 activities of essential oils were reported in [34,36,37], where one of the studies used a pseudotype SARS-CoV-2 focusing on its entry to the cell.

**Table 2.** Studies reporting molecular interactions of essential oils with COVID-19-related proteins selected based on systematic literature search strategies.

| Author, Year [Ref.] | Sample [a] | Major Compound * | In Vitro Assay/In Vivo Model | Outcome | Remarks |
|---|---|---|---|---|---|
| Ak Sakallı et al., 2022, [33] | Essential oils from *E. globulus* Labill. and *E. citriodora* Hook. | *E. globulus:* 1,8-Cineole (30.9%), α-pinene (11.4%), and β-pinene (11.4%) <br> *E. citriodora:* Citronellal (79.9%) | • ACE2 Inhibitor Screening Kit <br> • LOX activity inhibition assays | • At 20 µg/mL, *E. globulus* and *E. citriodora* essential oils inhibited 94.3% and 83.4% ACE2, respectively. <br> • At 20 µg/mL, *E. globulus* and *E. citriodora* essential oils inhibited 71.3%, and 91.4% 5-LOX, respectively. | Anti-SARS-CoV-2 activity is not determined. [33] |
| Demirci et al., 2021, [30] | Commercial menthol and essential oils from *M. arvensis* L., *M. citrata* L., and *M. spicata* L. | *M. arvensis*: Menthol (82%) <br> *M. citrata*: Menthone (22.2%), menthol (menthol) <br> *M. spicata*: Carvone (88.7%) | • ACE2 inhibition assay <br> • 5-LOX activity inhibition assays | • At 20 µg/mL, *M. arvensis*, *M. citrata*, and *M. spicata* essential oils inhibited 33.0 ± 0.13%, 22.1 ± 0.80%, and 73.2 ± 0.45% ACE2, respectively. <br> • At 5 µg/mL, menthol inhibited 99.8 ± 0.02% ACE2 <br> • At 20 µg/mL, *M. arvensis*, *M. citrata*, and *M. spicata* essential oils inhibited 84.5 ± 0.14%, 79.0 ± 0.12%, 70.1 ± 0.34% 5-LOX activities, respectively. <br> • At 5 µg/mL, menthol inhibited 79.9 ± 0.43% 5-LOX activity. | Anti-SARS-CoV-2 activity is not determined. |
| Demirci et al., 2022, [31] | Commercial 1.8-cineole and essential oil from *R. officinalis*. | *R. officinalis:* 1,8-cineole (62.7%), α-pinene (12.6%), and camphor (8.3%) | • ACE2 inhibition assay <br> • 5-LOX activity inhibition assays | • At 20 µg/mL, *R. officinalis* essential oil inhibited 20% ACE2. <br> • At 5 µg/mL, 1,8-cineole inhibited 89.2% ACE2. <br> • At 20 µg/mL, *R. officinalis* essential oil inhibited 81.13% 5-LOX activity. <br> • At 5 µg/mL, 1,8-cineole inhibited 37.17% 5-LOX activity. | Anti-SARS-CoV-2 activity is not determined. |
| Biltekin et al., 2022, [32] | *L. angustifolia, L. stoechas,* and *L. heterophylla* | *L. heterophylla*: Linalool (30.6%), linalool acetate (19.6), camphor (15%) and 1,8-cineole (11.3%) <br> *L. stoechas*: Camphor (54.7%) and α-fenchone (19.2%) <br> *L. angustifolia*: Camphor (17.9%), 1,8-cineole (12.3%), linalool (22.4%), and linalool acetate (19.2%). | • ACE2 inhibition assay <br> • 5-LOX activity inhibition assays | • At 20 µg/mL, *L. angustifolia*, *L. stoechas*, and *L. heterophylla* essential oils inhibited 25.4%, 34.1%, and 27.1% ACE2, respectively. <br> • At 5 µg/mL, Linalool and camphor inhibited 77.1% and 85.1% ACE2, respectively. <br> • At 20 µg/mL, *L. angustifolia*, *L. stoechas*, and *L. heterophylla* essential oils inhibited 79%, 49.1%, and 86.7% 5-LOX, respectively. <br> • At 5 µg/mL, Linalool and camphor inhibited 92% and 67.2% 5-LOX, respectively. | Anti-SARS-CoV-2 activity is not determined. |
| Asaad et al., 2022, [37] | *C. clementine* fruits were crushed in ethanol solution and added with water before filtration. The filtrate was partitioned with n-hexane to produce the essential oil. | *C. clementine*: Limonene (92.28%) | • Plaque reduction assay using SARS-CoV-2-infected Vero-E6 cells <br> • SARS-CoV-2-infected Huh-7 cells for TNF-α and IL-6 | • SI of 0.689 against SARS-CoV-2 replication. <br> • Amelioration of TNF-α and IL-6 after SARS-CoV-2 infection. <br> • Limonene binds spike protein of SARS-CoV-2 in silico. | The binding with SARS-CoV-2 spike protein is only observed by molecular docking. |

**Table 2.** *Cont.*

| Author, Year [Ref.] | Sample [a] | Major Compound * | In Vitro Assay/In Vivo Model | Outcome | Remarks |
|---|---|---|---|---|---|
| González-Maldonado et al., 2022, [34] | Essential oils: *G. sarmientoi, C. aurantium* L. var. amara, *M. frondosus, A. emarginata, E. globulus, L. alba, C. citratusv* Volatile compounds: β-Caryophyllene, Caryophyllene oxide, Linalool, Trans-anethole, S-Limonene, R-Limonene, cis-Verbenol, Guaiol, Macrophominol, Acetylphomolactone, Botryodiplodin, Asperline, Isoasperline | No compound identifications were carried out on the essential oil, but the commercial essential oil constituents (volatile compounds) were assayed directly. | • HIV-1 with SARS-CoV-2 spike protein (pseudotype virus) infected to HEK293-ACE2-expressing cells | • Antiviral activities of essential oils were found in the sample with high cytotoxicity, hence low SIs. <br> • *E. globulus* promotes viral entry. <br> • β-Caryophyllene specifically inhibits viral entry via spike protein with $62.10 \pm 10.31\%$ replication inhibition and 125 μg/mL MNTC. | Required confirmation using the whole SARS-CoV-2 |
| Kumar et al., 2020, [35] | Essential oils: *C. bergamia, P. nigrum, M. chamomilla, C. annum, C. winterianus, S. sclarea, C. sempervirens, C. valgare, E. globulus, F. vulgare, Boswellia* sp., *P. graveolens, Z. officinale, J. communis, K. ambigua, C. limon, L. officinalis, C. aurantifolia, L. cubeba, O. majorana, M. communis, C. aurantium, C. martinii, P. cablin, M. piperita, C. aurantium, C. camphora, R. officinalis, C. reticulata,* and *M. alternifolia.* Volatile compound: Citronellol, geraniol, neryl acetate, and limonene. | *C. limon*: Citronellol (27.1%), geraniol (21.4%), and neryl acetate (10.5%) *P. graveolens*: Limonene (73%) | • HT-29 cells | • Essential oils from *C. limon* and *P. graveolens* are the most active in downregulating ACE2 expression. <br> • Major components of *C. limon* and *P. graveolens* essential oils significantly downregulate expressions of ACE2 and TMPRSS2 mRNAs | Anti-SARS-CoV-2 activity is not determined. |
| Mohamed et al., 2022, [36] | Essential oil from *A. robusta* bark obtained through hydrodistillation | Tricyclene (11.89%), α-pinene (19.49%), d-camphene (7.13%), limonene (9.37%), trans-pinocarveol (4.95%), borneol (2.32%), α-phellandren-8-ol (2.51%), and α-terpineol (9.59%). | • SARS-CoV-2-infected Vero-E6 cells | • *A. robusta* bark essential oil has SI = 17.5 against SARS-CoV-2. <br> • Major components of the essential oil interact with M[pro], RdRp, and RBD of SARS-CoV-2 based on molecular docking. | The molecular interactions have not been confirmed in vitro/in vivo |

ACE2, angiotensin-converting enzyme 2; HIV-1, human immunodeficiency virus 1; LOX, lipoxygenase; MNTC, maximum non-toxic concentration; SARS-CoV-2, severe acute respiratory syndrome coronavirus 2; SI, selectivity index; TMPRSS2, transmembrane protease serine 2. [a] Otherwise stated, the samples were procured as commercial product. * Based on gas chromatography/mass spectrometry analysis.

None of the studies is considered adequate to evaluate the effects of essential oils against COVID-19-related proteins. Four studies only assayed essential oils based on enzyme inhibitory activity [30–33]. One study used pseudotype SARS-CoV-2 and required further confirmation with the complete virus [34]. Reductions of ACE2 and TMPRSS2 expressions were only observed based on a single cancer cell line (HT-29) [35]. While other studies solely used the molecular docking approach to determine the molecular interactions [36,37].

**Table 3.** Quality appraisal results of the included studies.

| Author, Year, Ref | Study Design Rationale | Reproducibility | Replication | Negative Control | Positive Control | Anti-SARS-CoV-2 Activity | Study Adequacy |
|---|---|---|---|---|---|---|---|
| Ak Sakallı et al., 2022, [33] | Yes | Yes | Yes | Yes | Yes/No [a] | No | No |
| Asaad et al., 2022, [37] | Yes | Yes | Yes | Yes | No | Yes | No |
| Demirci et al., 2021, [30] | Yes | Yes | Yes | Yes | Yes/No [a] | No | No |
| Demirci et al., 2022, [31] | Yes | Yes | Yes | Yes | Yes/No [a] | No | No |
| Biltekin et al., 2022, [32] | Yes | Yes | Yes | Yes | Yes/No [a] | No | No |
| González-Maldonado et al., 2022, [34] | Yes | Yes | Yes | Yes | No | Yes | No |
| Kumar et al., 2020, [35] | Yes | Yes | Yes | Yes | No | No | No |
| Mohamed et al., 2022, [36] | Yes | Yes | Yes | Yes | No | Yes | No |

[a] No positive control for ACE2 inhibition assay.

### 4.6. Non-Confirmed In Silico Studies

To obtain an insight into how the in silico studies have progressed to in vitro or in vivo studies in investigating the COVID-19-related molecular targets of essential oils, we included the non-confirmed studies, which have been presented in Table 4. There are as many as 14 studies performing the analysis of the anti-SARS-CoV-2 activity of the essential oils in silico. *Ammoides verticillate* (Desf.) Briq, *Stylosanthes guianensis*, *Copaifera langsdorffii*, *Matricaria recutita* L., *Ferula gummosa*, *Cucurma longa* L., family Lamiaceae, family Geraniaceae, *Melaleuca cajuput*, *Eucalyptus* sp., *Corymbia citrodora*, *Cymbopogon citratus* L, *Moringa oleifera*, and *Piper betle* are the studied plant resources of the essential oils [38–51]. Almost all the protein targets are those from SARS-CoV-2, with the addition of a human ACE2 receptor. It is worth noting that are more plants and essential oil compounds which have been investigated in silico but are not included herein. These plants and compounds have been thoroughly reviewed in a published systematic review that specifically included studies on essential oils with potential anti-SARS-CoV-2 activities investigated based on molecular docking approaches [22].

**Table 4.** SARS-CoV-2-related molecular targets used in molecular docking studies without in vitro confirmation.

| Author, Year, [Ref.] | Plant | Compound of Interest | Molecular Target |
|---|---|---|---|
| Abdelli et al., 2021, [38] | *Ammoides verticillate* (Desf.) Briq | Isothymol | ACE2 |
| Costa et al., 2022, [39] | *Stylosanthes guianensis Copaifera langsdorffii* | γ-Eudesmol, β-selinene | M$^{pro}$ |
| da Silva et al., 2020, [40] | *Matricaria recutita* L. | (*E,E*)-α-Farnesene, €-β-farnesene, (*E,E*)-farnesol | M$^{pro}$, endoribonuclease, ADP-ribose phosphatase, RdRp, spike RBD. ACE2 |
| Habibzadeh et al., 2022, [41] | *Ferula gummosa* | Δ-Cadinene, β-eudesmol, bulnesol | 3CLpro, Spike RBD, PLpro, RdRp |
| Kulkarni et al., 2020, [42] | Family Lamiaceae and Geraniaceae | Thymol, pulegone | Spike RBD |
| Mahomoodally et al., 2021, [43] | *Cucurma longa* L. | β-sesquiphellandrene, α-zingiberene | COVID-19 crystal structure |
| My et al., 2020, [44] | *Melaleuca cajuputi* | Guaiol and linanool | ACE2 |
| Panikar et al., 2021, [45] | *Eucalyptus globulus Corymbia citrodora* | 1.8-cineole | M$^{pro}$ |
| Sharma et al., 2020 [47] | *Eucalyptus* sp. | 1.8-cineole | M$^{pro}$ 3CL$^{pro}$ |
| Sharma et al., 2020 [46] | *Eucalyptus* sp. | Jensenone | 3CL$^{pro}$, M$^{pro}$ |
| Sharma et al., 2021 [52] | *Eucalyptus* sp. | Torquatone | Spike protein |
| Sharma et al., 2022 [49] | *Cymbopogon citratus* L. | Citral | Spike protein |
| Siddiqui et al., 2022, [50] | *Moringa oleifera* | 2-pyrrolidinone | Spike protein, ACE2 |
| Tu Quy PTA, 2022, [51] | *Piper betle* | Chavicol acetate, *trans*-Isoeugenol, Eugenol acetate | Spike protein |

## 5. Discussion

### 5.1. Targeting SARS-CoV-2-Related Proteins

Results from our systematic review suggest that essential oils could attenuate ACE2 and TMPRSS [30,31,33,35]. A study revealed that β-caryophyllene could selectively disrupt the binding of spike protein with ACE2 [34]. *C. clementine* essential oil had a selectivity index (SI) of 0.689 against SARS-CoV-2 replication, and its major constituent, limonene (92.28%), had a high binding affinity with SARS-CoV-2 spike protein in silico [37]. Studies using enzyme inhibition assay have shown that *E. globulus*, menthol, 1,8-cineole, and camphor have strong activity against human ACE2 [30–33].

It has been understood that the entry points of SARS-CoV-2 into host cells are the ACE2 receptor and TMPRSS2 receptor. Both proteins, ACE2 and TMPRSS2, are macromolecules with molecular sizes of around 100 kDA and 54 kDA, respectively [53,54]. Higher infectivity of SARS-CoV-2 as compared to the previous SARS-CoV is attributed to more compact and stable binding between the spike protein and ACE2 receptor [55]. The spike protein of SARS-CoV-2 is divided into two subunits, the globular head subunit (S1) and the stalk-like subunit (S2). There are four domains comprising S1 subunit, namely the receptor-binding domain (RBD) and *N*-terminal domain (NTD), as well as two structurally conserved subdomains [56]. The RBD part binds ACE2 on the host cell surface, and the S protein is cleaved during the membrane fusion step mediated by TMPRSS2 [17]. A representation of essential oils targeting proteins to inhibit viral entry and replication has been presented (Figure 2a,b).

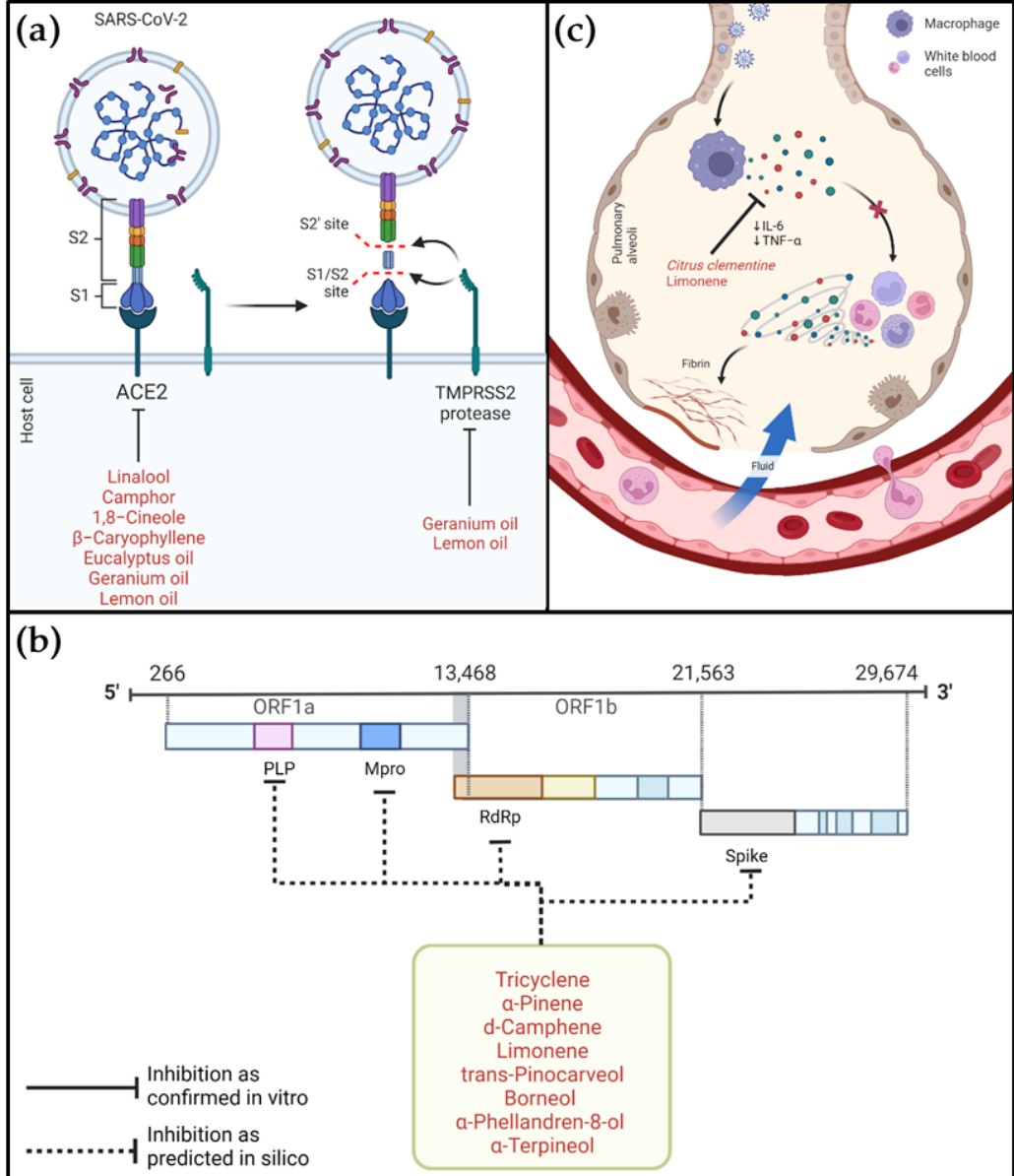

**Figure 2.** Mechanism of action of essential oil as anti-SARS-CoV-2 by blocking the ACE2 and TMPRSS2 as viral entry points (**a**) and inhibiting the viral proteins (**b**). Excessive release of cytokine by the immune system following SARS-CoV-2 infection causes a cytokine storm which could be treated by using Citrus clementine and limonene (**c**). Essential oils acting as anti-SARS-CoV-2 agents are presented in red.

Blocking the complex formation of spike protein ACE2 has been considered a therapeutic approach [6,57]. As a part of the Renin Angiotensin System (RAS), a complex system regulating systemic arterial pressure, ACE2 holds a major role in sustaining the balance of Angiotensin II (AngII) and Angiotensin-(1–7) (Ang (1–7)) levels [58]. Recent findings suggest that ACE2 is used by SARS-CoV-2 to enter host cells and subsequently downregulated by possible alteration of ACE2 RNA, leading to higher Ang II release that causes vasoconstriction, vasculitis, and inflammation [18]. Considering its double-edge nature, targeting ACE2 in treating COVID-19 should carefully consider possible dysregulation of the patient's RAS.

Drugs with inhibitory activity against TMPRSS2 may block cell surface fusion. Some clinically proven TMPRSS2 inhibitors are E-64d and camostat mesylate, which have been reported to attenuate the SARS-CoV-2 infection [59]. SARS-CoV-2 variants exert different

replication and fusion activities in TMPRSS2-expressing cells, such as in Omicron; the activities are lower compared to the Delta variant [60]. Therefore, the emergence of SARS-CoV-2 variants such as Omicron could affect the efficacy of this type of therapeutic approach.

Additionally, the essential oil constituents could bind to M$^{pro}$, RdRp, and RBD of SARS-CoV-2 in silico with positive anti-SARS-CoV-2 activity [36]. Papain-like protein (PLP) and M$^{pro}$ are involved in proteolytic cleavage of the viral polyproteins for producing a wide array of proteins essential in the replication of SARS-CoV-2 and even in the case of most positive-sense RNA viruses [61]. Proteolytic cleavage of the two coronavirus polyproteins generates the various viral proteins needed to form a replication complex, required for transcription and replication of the viral genome and subgenomic mRNAs. The key viral enzymes responsible in this regard are the papain-like (PLP) and main proteases (M$^{pro}$) [62]. Hence, blocking M$^{pro}$ by essential oil constituents could essentially lower the replication of the virus. Another essential protein is RdRp, a machinery for genome replication of positive-sense single-stranded RNA viruses [61]. Pharmaceutical agents targeting RdRp could inhibit viral replication by terminating RNA chain elongation, but they suffer from a lack of antiviral efficacy due to the proofreading activity of nidovirus [63]. Some approved antivirals that could escape the proofreading activity include remdesivir (delayed RNA chain termination) and molnupiravir (error catastrophe) [64,65]. It is important for the essential oils targeting RdRp to be investigated further to confirm their ability to escape viral proofreading.

### 5.2. Targeting Inflammatory Factors

The lethality of COVID-19 is often associated with cytokine storm, a hyperinflammatory condition formed after the body fails to downregulate the inflammatory reaction initiated by the immune response [66]. The release of chemokines, chemokines, and other inflammatory factors are responsible for systemic inflammation. In COVID-19 patients, interleukin(IL)-6, IL-7, and tumor necrosis factor (TNF) have been observed to be elevated [67]. Cytokine storm leads to the disease progression into acute respiratory distress syndrome (ARDS) and/or organ damage, which causes mortality among COVID-19 patients [68]. Essential oils have been proposed by multiple review articles as therapeutic agents for COVID-19, attributed to their abilities to downregulate the inflammatory factors [69,70].

In this light, the included studies herein have investigated their samples against pro-inflammatory factors. Essential oils from *Eucalyptus citriodora* Hook. could effectively inhibit the activity of 5-lipoxygenase (5-LOX) by >90% [33]. *R. officinalis* could inhibit 5-LOX as high as 81.13% [31]. Nonetheless, 1,8-cineole or eucalyptol, the major constituents of the foregoing plants, only inhibited 37.17% 5-LOX activity in another study [31]. *M. arvensis* exerted higher inhibition against 5-LOX (84.5 ± 0.14%) as compared with *M. citrate* (79.0 ± 0.12%) and *M. spicata* (70.1 ± 0.34%) [30]. Linalool was found to have a strong inhibitory activity against 5-LOX [32]. Nordihydroguaiaretic acid (NDGA), the positive control, always has an inhibition of >90% [30,31,33]. A review article suggested the hypothetical role of 5-LOX in COVID-19 pathophysiology [71]. The association of 5-LOX activities with COVID-19 severity is based on the fact that the expression of lipoxygenases occurs in monocytes, eosinophils, neutrophils, and macrophages [72]. 5-LOX converts free arachidonic acid into pro-inflammatory leukotrienes (LTs), and its metabolites trigger the release of pro-inflammatory cytokines and chemokines [73]. Nonetheless, there is little evidence available to suggest the involvement of 5-LOX in COVID-19 pathogenesis. There is indeed a non-peer-reviewed preprint suggesting that collective RNA sequences from deceased individuals with COVID-19 formed a unique network cluster of 5-lipoxygenase as one of the therapeutic targets of COVID-19 [74].

One of the included studies revealed the amelioration of TNF-$\alpha$ and IL-6 after SARS-CoV-2 infection in huh-7 cell lines by *C. clementine* essential oil containing 92.28% limonene [37]. As mentioned previously, these cytokines have been found to be elevated in COVID-19 patients [67,75]. Moreover, TNF-$\alpha$ and IL-6 are two of the three cytokine

triad members thought to be associated with long-term COVID-19 symptoms [76]. IL-6 blockers have been proposed as a therapeutic option for cytokine release syndrome, which is thought to play a major role in the pathology of COVID-19 [77]. The potential of limonene to reduce the inflammatory impact of COVID-19 has been long noted [78]. Through the mice model, studies have demonstrated the efficacy of limonene and its metabolites in reducing various cytokines [79,80]. It is worth mentioning that eucalyptus oil and its major constituent, eucalyptol, have been found effective in reducing pro-inflammatory cytokines released from monocytes and macrophages without interfering with their phagocytosis [81]. The mechanism of *C. clementine* essential oil and limonene in attenuating cytokine storm via IL-6 and TNF-$\alpha$ has been presented (Figure 2c). Hence, the in vitro model employed by [37] could be replicated for other essential oil, particularly for eucalyptus oil.

### 5.3. Other In Vitro Studies on Anti-SARS-CoV-2 Activity of Essential Oils

To complement the included studies, we present the studies reporting anti-SARS-CoV-2 activities of essential oils but without investigating their molecular targets (Table 5). Essential oil from *Nigella sativa* was found to yield SI of less than four [28], while *M. pulegium*, *Mentha microphylla*, *Mentha vilosa*, *Mentha thymifolia*, *Illicium verum*, *Syzygium aromaticum*, *C. limon*, and *Pelargonium graveolens* essential oils had SI higher than four against SARS-CoV-2 replication [26,29]. Interestingly, a study used pseudovirus of the delta variant [26]. Reduction of viral release as high as 80% was achieved by a study using a mixture of essential oils from *Thymbra capitata* (L.) Cav., *Salvia fruticosa* Mill., and *Origanum dictamnus* L. [27]. The volatile compounds predominantly found in the foregoing essential oils include 1,8-cineol, linalool, menthol, and limonene, in whose anti-SARS-CoV-2 molecular targets have been reported [30–33,37]. Taken altogether, these reports corroborate the suggestions that essential oil could be used in treating COVID-19, including those caused by variants of concern.

**Table 5.** Studies reporting anti-SARS-CoV-2 activity of essential oils but without molecular target investigation.

| Author, Year [Ref.] | Sample | Major Compound * | In Vitro Assay | Outcome |
|---|---|---|---|---|
| Zeljković et al., 2022 [29] | Essential oils: *Mentha* sp., *Micromeria thymifolia* (Scop.) Fritsch, and *Ziziphora clinopodioides* Lam | p-Cymene; thymol; carvacrol; limonene; 1,8-cineol; linalool; menthone; menthofuran; menthol; terpinene-4-ol; $\alpha$-terpineol; pulegone; and carvone | SARS-CoV-2-infected Vero 76 cells | *M. pulegium*, *M. microphylla*, *M. vilosa*, and *M. thymifolia* essential oils have SI => 13.47, 7.81, 9.27, and 6.73, respectively, against SARS-CoV-2 |
| Esharkawy et al., 2022 [28] | *Nigella satvia* | Thymoquinone 2,5-dihydroxy-para-cymene | SARS-CoV-2-infected Vero 76 cells | *N. sativa* essential oil has SI = 1.4 against SARS-CoV-2 |
| Lionis et al., 2021 [27] | *Thymbra capitata* (L.) Cav., *Salvia fruticosa* Mill., and *Origanum dictamnus* L. | Not reported | SARS-CoV-2-infected Vero 76 cells | Essential oils combination reduces the viral release up to >80% |
| Neto et al., 2022 [26] | *Syzygium aromaticum*, *Cymbopogon citratus*, *Citrus limon*, *Pelargonium graveolens*, *Origanum vulgare*, *Illicium verum*, and *Matricaria recutita* | (E)-Anetole, limonene, $\beta$-pinene, citronellol, and eugenol | SARS-CoV-2 delta pseudovirus infected to ACE2-expressing HeLa cells | *I. verum*, *S. aromaticum*, *C. limon*, and *P. graveolens* essential oils have SI > 4 (60, 4.4, 8.7, and 8.5, respectively) |

ACE2, angiotensin-converting enzyme 2; SARS-CoV-2, severe acute respiratory syndrome coronavirus 2; SI, selectivity index. * Based on gas chromatography/mass spectrometry analysis.

### 5.4. In Vitro Study Design

The study designs employed by the included studies herein are rather non-uniform, starting with the bioactivities measured up to the selection of the in vitro model. Cells used were also heterogenous (Huh-7, HT-29, and HEK293). Indeed, each study design has its own strengths and limitations, as summarized in Table 6. Researchers may opt

for one study design over another, depending on the research objective or even resource availability (especially for using complete SARS-CoV-2 culture, which requires a laboratory with Biosafety Level 3 standard [82]). To overcome this, the use of positive control could aid the comparability of the results. Unfortunately, all studies included herein did not use positive control contributing to the difficulty in comparing the results. Examples of the positive controls for the ACE2 spike protein and TMPRSS2 (based on 5$\alpha$-reductase activity) inhibition assays are monoclonal antibody AC384 and finasteride, respectively [83].

**Table 6.** Advantages and disadvantages of using various in vitro models as reported in the included studies.

| Method [Ref.] | Advantages | Disadvantages |
|---|---|---|
| Commercial assay kit [30,31,33] | • Easy to use<br>• Can confirm the specific interaction with the targeted protein | • Cannot be used for anti-SARS-CoV-2 activities<br>• Neglecting the physiological response |
| ACE2 expressing cancer cells [35] | • Unlimited biological material | • Results cannot be inferred for normal cell |
| Pseudotype virus [34] | • Low risk<br>• Not requiring high biosafety standard<br>• Enable investigations on specific protein<br>• Wide option of replication-permissive cell cultures | • Needs further confirmation using the real virus<br>• Complex preparation |
| SARS-CoV-2 virus [36,37] | • The closest mimicry to in vivo model<br>• Enable investigations of the related protein | • High risk<br>• Low replication permissiveness<br>• Influenced by other physiological factors |

Cell culture selection is an important consideration to yield results that are most representative of the human body. Differentiated primary airway epithelial cells are the most ideal to allow viral entry and replication of SARS-CoV-2 in a culture medium. It is due to the fact that ciliated and type 2 pneumocyte cells are the major target of SARS-CoV-2 infection [84]. However, the requirements of complex protocols and long duration for cell differentiation limit their utility in in vitro studies. Vero cells, isolated from African green monkeys, have been widely used in studying betacoronaviruses [85,86]. Other cells that are permissive to SARS-CoV-2 replication are hepatocellular carcinoma Huh-7 colorectal adenocarcinoma Caco-2 cell lines, where the replications have been found to be dependent on ACE2 [86,87]. One of the included studies herein employed a colorectal adenocarcinoma HT-29 cell line to investigate ACE2 inhibition activity [35]. The cancer cell line has been suggested to be non-permissive for SARS-CoV-2 replication [88], but it is thought to have relatively high ACE2 expression [89], hence the suitability for ACE2 inhibition assay. Moreover, high expression of TMPRSS2 is also reported for colorectal epithelial tissues of colorectal cancer patients [90]. However, ones should be aware of the differences in gene expression between cancer and normal cell lines, which have been suggested to be significant [91].

### 5.5. Comments on Molecular Docking Studies

Herein, we found 14 studies reporting the molecular interactions between essential oils and COVID-19-related proteins, but without in vitro/in vivo confirmation. *Eucalyptus* sp. and its major constituent, 1,8-cineole, are both reported in in silico and in vitro studies. Other essential oils were only reported in in silico studies. Of studied compounds, 1,8-cineole or eucalyptol has been studied in multiple reports [45,47] and has been confirmed in vitro to block ACE2 [31]. Despite strong binding affinities of *Eucalyptus* sp. essential oils against SARS-CoV-2 spike protein and proteases in silico [45–47], a study using pseudo-SARS-CoV-2 suggested the proviral activity of the essential oils [34]. Molecular docking indeed could predict the interaction of ligands (the volatile compounds) and protein, and in fact, it has been used widely in drug discovery research [92–94]. However, it provides

weak evidence because it neglects other factors influencing the bioactivity of the drug candidates. Moreover, the technique only presents ligand-protein interaction and not the effect of the interaction, stabilizing or inhibiting. Molecular docking completely neglects the inhibitory potential of the non-competitive inhibitors since they do not bind at the catalytic site. Previously, out of 55 plant metabolites, those with the highest binding affinities toward SARS-CoV-2 spike protein failed to exhibit the same activity in spike pro x ACE2 assays [95]. Taken altogether, studies using in silico method should further confirm the anti-SARS-CoV-2 activities of the essential oils in vitro or in vivo.

## 6. Conclusions and Recommendations

Several essential oils and their constituents could influence the activities of key proteins in SARS-CoV-2 entry and replication. β-Caryophyllene could inhibit the viral entry in the pseudo-SARS-CoV-2 model by specifically targeting the spike protein. *C. limon* and *P. graveolens* could inhibit SARS-CoV-2 entry by downregulating ACE2 and TMPRSS2 mRNA expressions. Eucalyptus oil, along with the volatile compounds (menthol, 1,8-cineole, and camphor), have been suggested to act as ACE2 blockers. The use of ACE2 inhibitors should be accompanied by strong caution on the balance of patients' RAS, which contributes to the aggravation of the inflammatory state. However, essential oils have the ability to downregulate the release of pro-inflammatory cytokines, chemokines, and other pro-inflammatory factors. Thus, the net clinical benefit of using essential oils in the management of COVID-19 requires further studies. Furthermore, it is important to consider the faith of the essential oil compounds in the body, which are converted into different metabolites. Future research should extend the investigation on the efficacy of these metabolites.

More studies on the molecular mechanisms of essential oil as anti-SARS-CoV-2 and the disease's therapeutical agents should be further continued using a proper and comprehensive study design. The use of whole SARS-CoV-2 remains the gold standard for antiviral candidate screening. However, the use of pseudovirus with SARS-CoV-2 spike protein may provide better suggestions on the specific antiviral activity (inhibiting the viral entry). The use of enzyme inhibition assay could be useful in revealing the molecular interaction, especially when accompanied by pharmacokinetic analysis. Because of the different SARS-CoV-2 replication permissiveness and genes (especially between normal and cell cancer), researchers may consider using different cell types for better conclusive results. Last but not least, researchers must be careful in making interpretations when using cancer cells as their model.

**Author Contributions:** Conceptualization, M.I.; methodology, M.I.; software, M.I.; validation, T.F.D., H.H., R.I. and B.G.; formal analysis, M.I. and B.G.; investigation, M.I., D.R.R., A.P. and B.G.; writing—original draft preparation, M.I., D.R.R. and A.P.; writing—review and editing, T.F.D., H.H., R.I. and B.G.; visualization, M.I.; supervision, H.H., R.I. and B.G.; funding acquisition, B.G. All authors have read and agreed to the published version of the manuscript.

**Funding:** The APC was funded by Universitas Syiah Kuala.

**Institutional Review Board Statement:** Not applicable.

**Informed Consent Statement:** Not applicable.

**Data Availability Statement:** All underlying data could be requested to the corresponding authors. PRISMA checklist is available on https://doi.org/10.6084/m9.figshare.21780170.v1 (accessed on 26 December 2020).

**Acknowledgments:** Authors wish to extend their gratitude to Universitas Syiah Kuala for any kinds of support given during the preparation of this manuscript. Authors appreciate the collaboration among researchers from the Innovative Sustainability Lab, PT. Biham Riset dan Edukasi—Indonesia. We also acknowledge the suggestions from the three reviewers (appeared anonymously during the peer-review stage) which have improved the quality of our present article.

**Conflicts of Interest:** The authors declare no conflict of interest.

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
