# Peer review of "Antiviral Molecular Targets of Essential Oils against SARS-CoV-2: A Systematic Review"

_scipharm, doi:10.3390/scipharm91010015_

Round 1

Reviewer 1 Report

In this systemic review article, the authors have explored the antiviral mechanistic details of essential oils for SARS-CoV-2. The manuscript is well-written. I have just one comment.

In line 150, the authors mentioned that studies (n=7) which did not report molecular targets, despite the anti-SARS-CoV-2 activities, were excluded as well [24-27]. Nevertheless, it is worth including these articles in this review for the following reasons.

a.     Though the authors have included articles demonstrating molecular targets for essential oils, it is important to note that antiviral activity was not tested for many of them. However, the excluded articles have information on the antiviral properties of essential oils. Hence, the excluded articles could serve as a complement to the articles that were included.

b.     For example, in reference 27, the authors tested antiviral activity for selected Lamiaceae essential oils but did not identify molecular targets. However, in reference 28, the authors identified molecular targets for Lamiaceae essential oils but not tested antiviral activity. 

Author Response

Dear Reviewer 1,

Thank you for availing the time to evaluate our submitted manuscript. Your comments are significant in enhancing the quality of our present work. Please find our responses below.

Comments from reviewer:

In this systemic review article, the authors have explored the antiviral mechanistic details of essential oils for SARS-CoV-2. The manuscript is well-written. I have just one comment.

In line 150, the authors mentioned that studies (n=7) which did not report molecular targets, despite the anti-SARS-CoV-2 activities, were excluded as well [24-27]. Nevertheless, it is worth including these articles in this review for the following reasons.

a.     Though the authors have included articles demonstrating molecular targets for essential oils, it is important to note that antiviral activity was not tested for many of them. However, the excluded articles have information on the antiviral properties of essential oils. Hence, the excluded articles could serve as a complement to the articles that were included.

 b.     For example, in reference 27, the authors tested antiviral activity for selected Lamiaceae essential oils but did not identify molecular targets. However, in reference 28, the authors identified molecular targets for Lamiaceae essential oils but not tested antiviral activity. 

Response from authors:

Thank you for your expert comments. Firstly, our apology, there was a typo, the number should be 4 instead of 7. Secondly, agree. Those studies could complement the included studies in this systematic review. However, since they do not answer our research question (molecular target), we cannot include the table in the results. In exchange, we add a new section to accommodate this issue. Please refer to the amendment below.

5.3 Other In Vitro Studies on Anti-SARS-CoV-2 Activity of Essential Oils

To complement the included studies, we present the studies reporting anti-SARS-CoV-2 activities of essential oils but without investigating their molecular targets (Table 5). Essential oil from Nigella sativa was found to yield SI less than 4 [28], while M. pulegium, M. microphylla, M. vilosa, M. thymifolia, I. verum, S. aromaticum, C. limon, and P. graveolens essential oils had SI higher than 4 against SARS-CoV-2 replication [26,29]. Interestingly, a study used pseudovirus of delta variant [26]. Reduction of viral release as high as 80% was achieved by a study using a mixture of essential oils from Thymbra capitata (L.) Cav., Salvia fruticosa Mill., and Origanum dictamnus L [27]. The volatile compounds predominantly found in the foregoing essential oils include 1,8-cineol, linalool, menthol, and limonene, in which their anti-SARS-CoV-2 molecular targets have been reported [30-34]. Taken altogether, these reports corroborate the suggestions that essential oil could be used in treating COVID-19, including those caused by variants of concern.

Table 5. Studies reporting anti-SARS-CoV-2 activity of essential oils but without molecular target investigation

Author, Year [Ref.]

Sample

Major compound*

In-vitro assay

Outcome

Zeljković et al., 2022 [29]

Essential oils:

Mentha sp., Micromeria thymifolia (Scop.) Fritsch, and Ziziphora clinopodioides Lam

p-Cymene; thymol; carvacrol; limonene; 1,8-cineol; linalool;  menthone; menthofuran; menthol; terpinene-4-ol; α-terpineol; pulegone; and carvone

SARS-CoV-2-infected Vero 76 cells

M. pulegium, M. microphylla, M. vilosa, and M. thymifolia essential oils have SI= >13.47, 7.81, 9.27, and 6.73, respectively, against SARS-CoV-2

Esharkawy et al., 2022 [28]

Nigella satvia

Thymoquinone 2,5-dihydroxy-para-cymene

SARS-CoV-2-infected Vero 76 cells

N. sativa essential oil has SI= 1.4 against SARS-CoV-2

Lionis et al., 2021 [27]

Thymbra capitata (L.) Cav., Salvia fruticosa Mill., and Origanum dictamnus L.

Not reported

SARS-CoV-2-infected Vero 76 cells

Essential oils combination reduces the viral release up to >80%

Neto et al., 2022 [26]

Syzygium aromaticum, Cymbopogon citratus, Citrus limon, Pelargonium graveolens, Origanum vulgare, Illicium verum, and Matricaria recutita

(E)‑Anetole, limonene, β‑pinene, citronellol, and eugenol

SARS‑CoV‑2 delta pseudovirus infected to ACE2-expressing

HeLa cells

I. verum, S. aromaticum, C. limon, and P. graveolens essential oils have SI>4 (60, 4.4, 8.7, and 8.5, respectively)

Reviewer 2 Report

The review entitled „Antiviral molecular targets of essential oils against SARS-CoV- 2” discloses the molecular mechanisms of essential oil as anti-SARS-CoV-2 agents. The authors have done a great job by literature searching on PubMed, Scopus, Scillit and CaPlus/SciFinder collecting the information until December 7th, 2022. The review is structured enough to make it easy to read. Authors have analyzed 93 resources from respected peer-reviewed journals concentrating predominantly their attention on such molecular targets as angiotensin converting enzyme 2 (ACE2), transmembrane serine protease 2 (TMPRSS2) and SARS-CoV-2 spike protein.

There are no additional corrections from my side regarding the scientific part of Review. Moreover, the Manuscript is written in good grammar manner that allows it acceptance in the present form.

Given the above, I highly recommend this Review for publication as it is.     

Author Response

Comments from reviewer:

The review entitled „Antiviral molecular targets of essential oils against SARS-CoV- 2” discloses the molecular mechanisms of essential oil as anti-SARS-CoV-2 agents. The authors have done a great job by literature searching on PubMed, Scopus, Scillit and CaPlus/SciFinder collecting the information until December 7th, 2022. The review is structured enough to make it easy to read. Authors have analyzed 93 resources from respected peer-reviewed journals concentrating predominantly their attention on such molecular targets as angiotensin converting enzyme 2 (ACE2), transmembrane serine protease 2 (TMPRSS2) and SARS-CoV-2 spike protein.

There are no additional corrections from my side regarding the scientific part of Review. Moreover, the Manuscript is written in good grammar manner that allows it acceptance in the present form.

Given the above, I highly recommend this Review for publication as it is.     

_____________________________________________________________________________________

Response from authors:

Firstly, we thank you for availing the time to review our manuscript. Secondly, we appreciate your positive feedback on our present work. We hope to receive the same impression once this manuscript has been published.

Reviewer 3 Report

The manuscript is good and can be acceptable after some revisions

1. The authors should revise the other studies of the EOs as antiviral against SARS-CoV- 2 and molecular targets such as 

10.3390/ph14111138

https://doi.org/10.3390/molecules27227893

and others

2. The compounds in Figure 2 should be revised according to the literature . I think the number of compounds will be increased

3. The plants in Table 4 should be revised according to the literature . I think the number of plants will be increased

Author Response

Dear Reviewer 3,

Thank you for availing the time to evaluate our submitted manuscript; we appreciate your acknowledgment of our work and give some suggestions. In this opportunity, allow us to discuss the stated matters with you. Please find our responses below.

Comments from reviewer:

The manuscript is good and can be acceptable after some revisions

1. The authors should revise the other studies of the EOs as antiviral against SARS-CoV- 2 and molecular targets such as 

10.3390/ph14111138

https://doi.org/10.3390/molecules27227893

and others

Response from authors:

We appreciate your concern. However, the reviewer should realize that this present study is a systematic review so we are bound to the inclusion and exclusion criteria stated in the protocols. The systematic review suggested by the reviewer is not in line with our objective to identify the molecular target (not only the anti-SARS-CoV-2). But there is a possibility of the same concerns raised by the readers. Hence, we made an amendment to the original text to depict the distinction between our study with those suggested by the reviewer:

Indeed, there have been several reviews emphasizing the anti-SARS-CoV-2 potential of essential oils [21,22]. One of which was a narrative review [21], while another was a systematic review [22]. However, this is the first systematic review focusing on the molecular targets of essential oils while acting as anti-SARS-CoV-2. More specifically, this review performed a quality appraisal analysis on the included studies, adding its novelty aspect.

Comments from reviewer:

2. The compounds in Figure 2 should be revised according to the literature . I think the number of compounds will be increased

Response from authors:

We appreciate your concern. Nonetheless, the reviewer should keep in mind that the Figure 2 is established based on the included studies (Table 2), selected through a pre-planned protocol of systematic review itself. Addition of compounds that are not from the included studies based on the protocol employed would violate the protocol itself. Hence, with all your respect, we choose to maintain the Figure 2 as is.

Comments from reviewer:

3. The plants in Table 4 should be revised according to the literature. I think the number of plants will be increased

Response from authors:

Thank you for your remark. We agree that there supposedly more plants and compounds to be reported. However, since the protocol of this present systematic review was design to only include those which have been confirmed in vitro or in vivo (for in silico studies), it is then expected that we could only include a limited number of in silico-investigated anti-SARS-CoV-2 essential oils. However, we realized on the importance of this issue. Hence, an additional remark has been amended to the original text:

Please refer to this part:

It is worth noting that are more plants and essential oil compounds which have been investigated in silico which are not included herein. These plants and compounds have been thoroughly reviewed in a published systematic review that specifically included essential oils with potential anti-SARS-CoV-2 activities investigated based on molecular docking approaches [50].

Round 2

Reviewer 3 Report

I think the authors did all the required revisions and it can be accepted in the present form